# Mortality Risk Factors for Individuals Experiencing Homelessness in Catalonia (Spain): A 10-Year Retrospective Cohort Study

**DOI:** 10.3390/ijerph18041762

**Published:** 2021-02-11

**Authors:** Fran Calvo, Oriol Turró-Garriga, Carles Fàbregas, Rebeca Alfranca, Anna Calvet, Mercè Salvans, Cristina Giralt, Sandra Castillejos, Mercè Rived-Ocaña, Paula Calvo, Paz Castillo, Josep Garre-Olmo, Xavier Carbonell

**Affiliations:** 1Departament de Pedagogia, Institut de Recerca Sobre Qualitat de Vida, Universitat de Girona, 17004 Girona, Spain; sandracastillejoslarruy@gmail.com; 2Department of Quality Assessment, Evaluation and Research, Health and Community Foundation, 08010 Barcelona, Spain; 3Ageing, Disability and Health Research Group of Girona Biomedical Research Institute [IdIBGi], 17190 Salt, Spain; oriol.turro@ias.cat (O.T.-G.); josep.garre@udg.edu (J.G.-O.); 4Centre d’Acolliment i Serveis Socials “la Sopa”, Ajuntament de Girona, 17004 Girona, Spain; cfabregas@ajgirona.cat; 5Centro de Atención Primaria Santa Clara, Institut Català de la Salut, 17004 Girona, Spain; ralfranca.girona.ics@gencat.cat (R.A.); msalvans.girona.ics@gencat.cat (M.S.); 6Unitat d’Aguts, Institut d’Assistència Sanitària, 17190 Salt, Spain; anna.calvet@ias.cat (A.C.); mariap.castillo@ias.cat (P.C.); 7Centro de Atención Primaria Blanes, Institut Català de la Salut, 17300 Blanes, Spain; cgiralt.girona.ics@gencat.cat; 8Escola Universitària d’Infermeria i Teràpia Ocupacional, EUIT, Universitat Autònoma de Barcelona, UAB, 08221 Terrassa, Spain; mercerived@euit.fdsll.cat; 9Department of Medical Sciences, School of Medicine, University of Girona, 17004 Girona, Spain; paulacb15@gmail.com; 10Facultat de Psicologia, Ciències de l’Educació i l’Esport Blanquerna, Universitat Ramon Llull, 08022 Barcelona, Spain; xaviercs@blanquerna.url.edu

**Keywords:** homelessness, mortality, infectious disease, immigration, mental health, drug use disorder, alcohol use disorder, type 2 diabetes

## Abstract

(1) Background: Current evidence suggests that mortality is considerably higher in individuals experiencing homelessness. The aim of this study was to analyze the mortality rate and the mortality risk factors in a sample of individuals experiencing homelessness in the city of Girona over a ten-year period. (2) Methods: We retrospectively examined the outcomes of 475 people experiencing homelessness with the available clinical and social data. Our sample was comprised of 84.4% men and 51.8% foreign-born people. Cox’s proportional hazard models were used to identify mortality risk factors between origin groups. (3) Results: 60 people died during the ten-year period. The average age of death was 49.1 years. After adjusting for demographic characteristics and the duration of homelessness, the risk factors for mortality were origin (people born in Spain) (HR = 4.34; 95% CI = 1.89–10.0), type 2 diabetes (HR = 2.9; 95% CI = 1.62–5.30), alcohol use disorder (HR = 1.9; 95% CI = 1.12–3.29), and infectious diseases (HR = 1.6; 95% CI = 1.09–2.39). Our results show a high prevalence of infectious and chronic diseases. Type 2 diabetes emerges as an important risk factor in homelessness. The average age of death of individuals experiencing homelessness was significantly lower than the average age of death in the general population (which is greater than 80 years). (4) Conclusions: Foreign-born homeless people were generally younger and healthier than Spanish-born homeless people. Chronic diseases were controlled better in Spanish-born people, but this group showed an increased risk of mortality.

## 1. Introduction

Current evidence suggests that mortality is considerably greater in individuals experiencing homelessness than in the general population [1,2,3]. The excess risk is most evident in younger homeless people and, according to some studies, in women [4]. The standardized mortality ratio reported is typically, depending on the study, two to five times higher than in the general population [5]. The life expectancy of individuals experiencing homelessness is several years shorter, depending on the age and gender of the analyzed group. For instance, researchers in a Canadian study based on data from 2008–2010 observed that the average life span of a homeless person amounts to only 49 years, 28 years shorter than their housed counterparts [6]. Data obtained in the United States [7], France [8], Denmark [9], and Scotland [10] are quite similar, suggesting the universality of this phenomenon, at least in Western countries.

Among the causes of early mortality are infectious diseases such as human immunodeficiency virus (HIV), tuberculosis, ischemic heart disease, substance misuse, and external factors (unintentional injuries, suicides, homicides, and poisoning) [11]. These are in addition to high exposure to risk factors, including alcohol, tobacco, illicit drugs, and mental disorders [3]. Moreover, homelessness is an independent risk factor for death [10].

In addition to its high human cost, homelessness also results in the massive expenditure of public funds. It is estimated that the total annual cost of housing exclusion in Europe is 194 billion euros. Effective homelessness policies would reduce both human and financial costs [6,7,8,12,13,14,15].

Marginalized populations are poorly recorded in key data sources, which leave numerous gaps on the subject of homelessness [5,16]. Additionally, the literature tends to treat the settlement, health, and homelessness of foreign-born people as separate agendas [4,17]. Despite growing evidence that foreign-born people make up an increasing proportion of the homeless population, very few studies analyze the issue in terms of health and mortality [18,19]. These gaps must be filled in order to design successful interventions and achieve better health and financial outcomes.

Finally, not all studies define homelessness in the same way, and the categories used to classify homeless individuals depend on political positions in each country [20]. In Spain, the official definition of a homeless person—which is very restrictive—refers to an adult who has used services specifically for homeless people in the week preceding the count of homeless people in municipalities of more than 20,000 inhabitants [21]. We follow the Fèdération Européene d’Associations Nationales Travaillant avec les Sans-Abri (FEANTSA) and the European Observatory on Homelessness (EOH) in using the European Typology of Homelessness and Housing Exclusion (ETHOS) [22]. This typology is derived from the physical, social, and legal interpretation of what a “home” is; the term “homeless” includes not only people who live outdoors or in specific centers, but also those who live in insecure or inadequate housing.

Given the lack of scientific data on the impact of homelessness on health and mortality in Spain, we aimed to determine the mortality rate and identify the mortality risk factors of a cohort of individuals experiencing homelessness. In doing so, we distinguished between Spanish-born and foreign-born people. Spain is one of the main ports of entry for immigrants from Africa and Eastern Europe [23]. Our analysis sheds light on the characteristics of immigrants at risk of social exclusion and how they are related to health.

## 2. Materials and Methods

### 2.1. Design

Retrospective cohort study.

### 2.2. Population and Sample

The target population was individuals experiencing homelessness in the city of Girona (Catalonia, Spain) who were assisted through social services aimed specifically at homeless people. We defined homelessness according to the European Typology of Homelessness and Housing Exclusion [22]. The sample of this study was the 475 individuals recorded in 2006 by the city of Girona as persons experiencing homelessness and for whom clinical and social information was available.

### 2.3. Procedure

We reviewed the health and social service records of members of the sample from (a) the Catalan public health system, (b) Catalan mental health services, and (c) the primary municipal social services of the city of Girona (food and temporary overnight accommodations, short- and mid-term housing, assistance by social services).

### 2.4. Sample Information

Sociodemographic: Gender, date of birth, education level, origin (foreign-born vs. Spanish-born), and self-reported criminal record. We defined foreign-born people as those born in a country other than Spain. Origin is an invariable status pertaining to the country of birth of the individual, regardless of the migratory status of the individual’s parents or grandparents [24,25].

Survival status: Number of days of homelessness was calculated for each person assisted from 1 January 2006 to death or to 31 December 2015. To calculate this figure, we checked the health and social services records for each person and used the information provided therein to estimate the number of days spent homeless during each year. We also estimated the amount of time the person spent in Girona.

Clinical information: The clinical information was categorized into three groups: infectious diseases, chronic diseases, and psychiatric disorders. The infectious diseases registered were HIV, hepatitis C, and tuberculosis. High blood pressure, type 2 diabetes, and chronic obstructive pulmonary disease were coded as chronic diseases following the 10th version of the International Diseases Classification. Finally, psychiatric disorders included depressive disorders, anxiety disorders, schizophrenia spectrum, and other psychotic disorders, adjustment disorders, personality disorders, and substance and alcohol use disorder [26]. Survival was categorized as a dichotomous variable (living/deceased).

Statistical analysis: The clinical and demographic characteristics of the subjects were described. Comparison analysis was done between deceased and living using Chi-square tests for categorical variables and the Student *t*-test for continuous variables. Kaplan-Meier survival curves were constructed to estimate the survival distribution, and the log-rank test was used to compare survival years between differentiated groups. The hazard ratio for mortality between homeless groups was estimated by fitting a multivariate Cox proportional hazard model.

We adjusted the raw model for age, gender, and days of homelessness in order to quantify the effect of the sociodemographic variables. Further models were computed to quantify the effect of infectious and chronic diseases and psychiatric disorders. Results are expressed as absolute numbers and percentages, means, standard deviations, interquartile ranges, hazard ratios, and 95% confidence intervals (CI). Statistical tests were considered to be significant with a *p*-value < 0.05. With a sample size of 475 subjects, a type I error rate of 5% and an overall probability of the event equal to 15%, the statistical power to detect a hazard ratio equal or higher to 2.0 in a multivariate model including four covariates is 83%.

## 3. Results

The mean age of the study cohort (*n* = 475) was 33.4 years (SD = 12.1) for men (84.4% of the sample) and 33.3 years (SD = 12.5) for women. One hundred individuals (67.1%) had criminal records and, among them, 26 had been in jail. More than half of the sample were foreign-born (51.8%). The biggest group of foreign-born individuals came from Morocco (*n* = 134), and the others (*n* = 108) came from more than thirty different countries. More information about the sample and homelessness-chronicity was published by Calvo et al. (2020).

With regard to infectious diseases, 13.7% of the cohort had hepatitis C, 6.5% HIV and 4.6% tuberculosis. Among chronic diseases, the most common was high blood pressure (15.4%), followed by type 2 diabetes (10.1%) and chronic obstructive pulmonary disease (8.0%). Regarding registered mental disorders, anxiety disorder was the most frequently recorded diagnosis (23.4%). Schizophrenia spectrum and other psychotic disorders were presented by 10.9% of the sample, depressive disorders by 10.3%, and personality disorders by 6.1%. In terms of substance use, alcohol use disorder was recorded in one-third of all cases, more than double the number of opioid users (13.1%) and cocaine users (12.4%). Table 1 shows more details about the overall sample and the comparative analysis by country of birth.

We examined the records of the cohort for a 10-year period. The mean time spent in the cohort was 8.1 years. During the study period, sixty people died (12.6%); among them, fifty-one were men (85.0%). Table 2 shows mortality by demographic and clinical data.

Significant statistical differences were found in vital status by origin (foreign-born vs. Spanish-born) (*χ*^2^ = 38.9; *d f* = 1; *p <* 0.001). Concerning mortality among infectious diseases, hepatitis C must be highlighted as a risk factor with 23.3% of deaths (*p* < 0.05). In terms of chronic diseases, significant statistical differences in mortality were found for all variables. Spanish-born people showed more personality disorders (10% vs. 2.4% of foreign-born people) and alcohol use (39.3% vs. 22.8%); no other differences in mental disorders and drug use were reported.

Table 3 shows the multivariate models adjusted for gender, age at baseline, and days of homelessness to identify clinical risk factors for mortality.

The multivariate analysis carried out with all variables is presented in Table 4. In Table 5 we can see the model that best explains mortality. Age (OR = 1.06; CI 95% 1.03–1.08), infectious disease (OR = 1.61; CI 95% 1.09–2.39), type 2 diabetes (OR = 2.93; IC 95% 1.62–5.30), alcohol abuse (OR = 1.92; CI95% 1.12–3.29) and, especially, being Spanish-born (OR = 4.3; CI 95% 1.9–10.0), were the factors associated with mortality.

Figure 1 illustrates the adjusted Cox regression comparing mortality between foreign-born and Spanish-born individuals.

## 4. Discussion

This study focuses on the mortality rate and mortality risk factors of people experiencing homelessness in Catalonia. The results show that the average age at death was 49 and five variables were associated with mortality: age, origin (Spanish-born), hepatitis C, type 2 diabetes, and alcohol abuse.

### 4.1. Main Mortality Figures

Several studies carried out from the 1970s to now show that homelessness and extreme poverty are independent risk factors for death from specific causes [10]. Homelessness increases the risk of mortality from 3 to 13 times in comparison with the general population around the world and specifically in Europe [2,3,9,27]. In France, the mean age at death was 49—the same as in our research. The leading causes of death among the homeless were external causes (20%), neoplasms (18%), diseases of the circulatory system (11%), diseases of the digestive system (7%), mental and behavior disorders (7%), and other causes (9%). It is important to consider that while age appears as one of the major risk factors of mortality, the mean age at death was significantly lower than the average of the general population [28,29].

The prevalence of infectious diseases was higher in individuals experiencing homelessness than in the general population of Catalonia (hepatitis C was 29 times more prevalent in the homeless population than in the general population, HIV 31 times more prevalent and tuberculosis 33 times more prevalent). The prevalence of type 2 diabetes and chronic obstructive pulmonary disease was also higher than in the general population [30]. It is important to highlight the high risk of mortality related to diabetes mellitus. People with diabetes who also experience homelessness often have difficulties with self-care. They suffer food insecurity and face financial barriers to accessing medicines. Chronic complications (kidney disease, myocardial infarction, amputation) are much more frequent in this group than in the general population [31].

### 4.2. The Effect of Origin

Homeless people who are foreign-born may have language, legal, and social barriers to accessing the health system, meaning that they may be less likely to get their illness under control. In this study, chronic diseases, mental disorders, and drug or alcohol abuse differ by origin. However, hepatitis C and type 2 diabetes were significantly more prevalent in Spanish-born homeless people, even though both were an independent risk factor for mortality.

Social inequality and cultural differences could have influenced these disease patterns [32]. For example, the country of birth among migrants could have influenced the possible cause of death [33]. Circulatory disease and diabetes are heavily stratified by socioeconomic position (e.g., nutrition or living conditions) and there is a high prevalence of poor nutrition and sedentarism—risk factors for coronary disease—in many foreign-born groups [34]. A Swedish study concluded that immigrants have an increased incidence of first acute myocardial infarction which persists several years after immigration and is not explained by socioeconomic differences [35].

Several researchers have explored the question of why there are differences between foreign-born and locally born homeless people [36,37,38]. One possibility is the “healthy immigrant effect” theory [39], which suggests that recent foreign-born are generally healthier than non-migrant populations, even though they frequently have lower socioeconomic status and less access to healthcare. Another explanation is immigrant self-selection. Migration is a difficult process, and for this reason, immigrant people usually are younger, healthier, and better prepared to confront stressful situations than their compatriots who do not migrate. Finally, some have posited that the difference is due to “cultural buffering” in immigrant groups. According to this theory, immigrant communities are protected by their social, familial, and religious networks, leading them to have healthier lifestyles and consume fewer cigarettes, alcohol, or illegal substances [27,40].

Some limitations should be taken into account. First, a large number of cohort members had no health records. The reason could be that they did not need health care or because there were some unidentified barriers to accessing services. Second, because clinical information was obtained once the person was already diagnosed in a public service, the prevalence is likely lower than we would have identified in a prospective study. In this sense, the homeless population is a very difficult group to access (especially when considering the broader definition) and to follow. The focus of the current study was variable mortality, which characteristically depends on an endless list of interrelated factors. Moreover, health problems were addressed by their absence or presence, without taking into account their severity. Finally, 10 years is a long period for a cohort study in the homeless population in comparison to other studies, but it is not long enough to study mortality linked to the mean age of individuals.

In conclusion, this study shows that the differences in health and mortality among our cohort of homeless people depended on several factors, most importantly origin (foreign-born or Spanish-born). Our study suggests that the “healthy immigrant effect,” despite the diversity of migrant populations across continents, appears to be present in Europe and seems to apply not only to immigrants living in secure conditions but also to highly disadvantaged and marginalized groups such as the homeless. Forthcoming studies should explore the differences in needs among foreign-born and locally born homeless populations and identify effective actions for each group.

We are currently collecting data from the same cohort to extend the timeframe through 2020 (15 years), allowing us to examine the longer-term effects of homelessness on health and mortality and offer perspectives on the COVID-19 pandemic. Moving forward, the creation of a national common dataset recording homeless health and demographic characteristics would be very useful. In this way, a more effective follow-up could be carried out to support homeless people’s health. This outreach would prioritize chronic illnesses with a higher risk of mortality, especially among people whose contact and follow-up with health services are low. Finally, it is important to put more social outreach teams on the street from different professional areas including nursing, medicine, psychiatry, language services, social work, and psychology to try to put these services in touch with homeless people, rather than expecting homeless people to make the approach.

## Figures and Tables

**Figure 1 ijerph-18-01762-f001:**
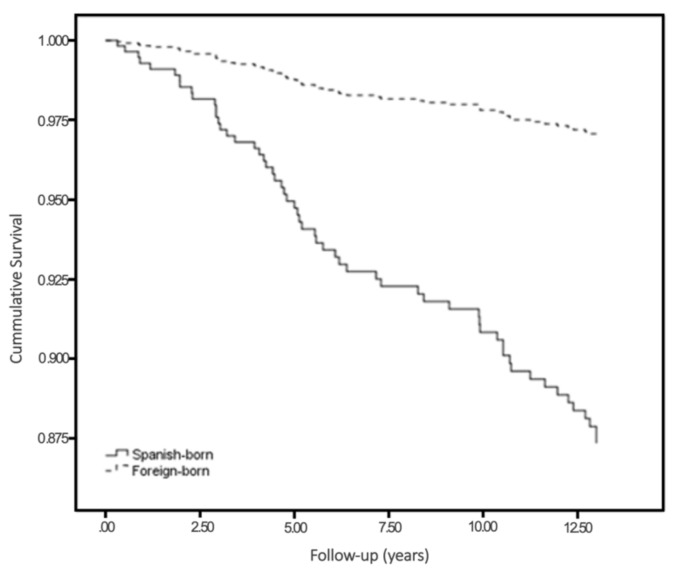
Cumulative survival adjusted model, by origin.

**Table 1 ijerph-18-01762-t001:** Demographic and clinical characteristics of the study participants.

Characteristics	Total (*n* = 475)
Age, mean (SD)	39.4 (12.1)
Days homeless, median (IQR)	119 (570)
Deceased, *n* (%) *	60 (12.8)
Country of birth, *n* (%)	
Catalonia	91 (19.2)
Spain **	142 (29.9)
Morocco	134 (28.2)
Others	108 (22.7)
Infectious diseases, *n* (%)	
AIDS	31 (6.5)
HCVI	65 (13.7)
TBI	22 (4.6)
Chronic diseases, *n* (%)	
HBP	73 (15.4)
DM	48 (10.1)
COPD	38 (8.0)
CVD	72 (8.4)
Mental disorders, *n* (%)	
Depressive disorder	49 (10.3)
Psychotic disorder	52 (10.9)
Personality disorder	29 (6.1)
Substance abuse, *n* (%)	
Alcohol	148 (31.2)
Cocaine	59 (12.4)
Opioids	62 (13.1)
Cannabis	34 (7.2)

* 6 missing values; SD: Standard deviation; IQR: Interquartile range; AIDS: Acquired immunodeficiency syndrome; HCVI: Hepatitis C virus infection; TBI: Tuberculosis infection; HBP: High blood pressure; DM: Diabetes mellitus; COPD: Chronic obstructive pulmonary disease; CVD: Cardiovascular disease. ** Not including data from Catalonia.

**Table 2 ijerph-18-01762-t002:** Demographic and clinical characteristics according to vital status at the end of the study period.

Characteristics	Global (*n* = 475)	Spanish-Born (*n* = 242)	Foreign-Born (*n* = 246)
Age, mean (SD)	39.4 (12.1)	43.8 (12.5)	34.9 (9.3)
Days homelessness, median (IQR)	119 (570)	722.6 (1147.1)	454.9 (897.5)
Deceased *, *n* (%)	60 (12.8)	52 (22.9)	8 (3.3)
Infectious diseases, *n* (%)			
AIDS	31 (6.5)	21 (9.2)	10 (4.1)
HCVI	65 (13.7)	51 (22.3)	14 (5.7)
TBI	22 (4.6)	12 (5.2)	10 (4.1)
Chronic diseases, *n* (%)			
HBP	73 (15.4)	51 (22.3)	22 (8.9)
DM	48 (10.1)	28 (12.2)	20 (8.1)
COPD	38 (8.0)	30 (13.1)	8 (3.3)
CVD	72 (8.4)	29 (12.7)	11 (4.5)
Mental disorders, *n* (%)			
Depressive disorder	49 (10.3)	32 (14.0)	17 (6.9)
Psychotic disorder	52 (10.9)	13 (5.7)	12 (4.9)
Personality disorder	29 (6.1)	23 (10.0)	6 (2.4)
Substance abuse, *n* (%)			
Alcohol	148 (31.2)	90 (39.3)	56 (22.8)
Cocaine	59 (12.4)	35 (15.3)	16 (6.5)
Opioids	62 (13.1)	46 (20.1)	16 (6.5)
Cannabis	34 (7.2)	14 (6.1)	13 (5.3)

* 6 missing values; SD: Standard deviation; IQR: Interquartile range; AIDS: Acquired immunodeficiency syndrome; HCVI: Hepatitis C virus infection; TBI: Tuberculosis infection; HBP: High blood pressure; DM: Diabetes mellitus; COPD: Chronic obstructive pulmonary disease; CVD: Cardiovascular disease.

**Table 3 ijerph-18-01762-t003:** Demographic and clinical characteristics according to vital status at the end of the study.

Characteristics	Living (*n* = 409)	Deceased (*n* = 60)
Age, mean (SD)	38.0 (11.1)	49.1 (14.3)
Days homeless, median (IQR)	113 (484)	225 (795)
Origin, *n* (%)		
Spanish-born	175 (77.1)	52 (22.9)
Foreign-born	234 (96.7)	8 (3.3)
Infectious diseases, *n* (%)		
AIDS	22 (5.4)	5 (8.3)
HCVI	47 (11.5)	14 (23.3)
TBI	13 (3.2)	5 (8.3)
Chronic diseases, *n* (%)		
HBP	53 (13.0)	16 (26.7)
DM	28 (6.8)	16 (26.7)
COPD	24 (5.9)	10 (16.7)
CVD	21 (5.1)	15 (25.0)
Mental disorders, *n* (%)		
Depressive disorder	39 (9.5)	10 (16.7)
Psychotic disorder	45 (11.0)	7 (11.7)
Personality disorder	20 (4.9)	9 (15.0)
Substance abuse, *n* (%)		
Alcohol	117 (28.6)	30 (50.0)
Cocaine	51 (12.5)	7 (11.7)
Opioids	50 (12.2)	10 (16.7)
Cannabis	32 (7.8)	2 (3.3)

SD: Standard deviation; IQR: Interquartile range; AIDS: Acquired immunodeficiency syndrome; HCVI: Hepatitis C virus infection; TBI: Tuberculosis infection; HBP: High blood pressure; DM: Diabetes mellitus; COPD: Chronic obstructive pulmonary disease; CVD: Cardiovascular disease.

**Table 4 ijerph-18-01762-t004:** Relative risk regression models for mortality for each cluster of elements.

Model 1 *		Model 2 *		Model 3 *		Model 4 *	
	HR (95% CI)		HR (95% CI)		HR (95% CI)		HR (95% CI)
AIDS	1.16 (0.42–3.17)	HBP	0.85 (0.43–1.69)	Depressive disorder	1.40 (0.67–2.93)	Alcohol	2.18 (1.26–3.77)
HCVI	1.7 (0.87–3.56)	DM	2.80 (1.50–5.25)	Psychotic disorder	0.89 (0.39–3.95)	Cocaine	0.90 (0.36–2.28)
TBI	1.83 (0.70–4.74)	COPD	1.11 (0.51–2.38)	Personality disorder	1.83 (0.84–3.91)	Opioids	1.80 (0.80–4.05)
		CVD	1.79 (0.86–3.71)			Cannabis	0.39 (0.08–1.75)

* Adjusted for age, gender, and number of days homeless; HR: Hazard ratio; CI: 95% confidence interval; AIDS: Acquired immunodeficiency syndrome; HCVI: Hepatitis C virus infection; TBI: Tuberculosis infection; HBP: High blood pressure; DM: Diabetes mellitus; COPD: Chronic obstructive pulmonary disease; CVD: Cardiovascular disease.

**Table 5 ijerph-18-01762-t005:** Model that best explains mortality.

	HR (95% CI)
Age	1.06 (1.03–1.08)
Gender	
Male	1
Female	0.81 (0.37–1.77)
Days homeless	1.0 (0.99–1.01)
Country of origin	
Foreign-born	1
Spanish-born	4.34 (1.89–10.0)
Infectious diseases	1.61 (1.09–2.39)
DM2	2.93 (1.62–5.30)
Mental disorders	1.16 (0.86–1.56)
Alcohol abuse	1.92 (1.12–3.29)

## Data Availability

The data presented in this study are available on request from the corresponding author. The data are not publicly available due to ethics and privacy requirements.

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
