# Peer review of "Mortality Risk Factors for Individuals Experiencing Homelessness in Catalonia (Spain): A 10-Year Retrospective Cohort Study"

_ijerph, 2021, doi:10.3390/ijerph18041762_

Round 1
Reviewer 1 Report
This is a very important topic, and there is indeed relatively very little research on the issue.
Studying the homeless, the house, national, and immigrants at the same time and comparing health outcomes is something that I have done in the US.S and called for others to do for the last ten years in the US. Glad to see this work in Catalonia, particularly starting in 2006 and having a longitudinal component. This is a great accomplishment.
Using a more expansive category of homeless (ETHOS) than is done in the local homeless census is a STRENGHT of the paper. A small limitation is that it seems to include only people using services.
Line 233 should be AND cultural, but most importantly, there is no need to mention genetic differences as an explanation of health disparities. That is unintentional a racist statement, scientifically speaking, genetic differences within Catalonia and Spanish-born people could be as large or larger than those between the foreign-born. This is a minor issue that can be easily solved by deleting the terms biology or genetic. This goes beyond the data and design of the study so this is not a fatal flaw. Showing the differences is in itself a contribution. I hope I am clear and not offensive. This related to the findings of the Human Genome Project that we share 99% of our DNA and “race” is more of a social construction than a biological fact. This is important because leaving biology as an explanation here plays to a very common belief among the population but one not backed up by the biology literature and with important eugenic implications. I would be happy to elaborate if unclear.
Otherwise, this is an important study and welcome addition to the literature.
Castañeda, Ernesto, Blaine Smith, and Emma Vetter. 2020. “Hispanic Health Disparities and Housing: Comparing Measured and Self-Reported Health Metrics among Housed and Homeless Latin Individuals.” Journal of Migration and Health. 1(1-2). 100008.
Castañeda, Ernesto, and Curtis Smith. 2020. “Sick enough? Mental Illness and Service Eligibility for Homeless Individuals at the Border.” Social Sciences. Volume 9, Number 8, 23 pp.
Smith, Curtis and Ernesto Castañeda. 2019. “Improving Homeless Point-In-Time Counts: Uncovering the Marginally Housed.” Social Currents. Volume 6, Number 2, pp. 91–104.
Author Response
Dear reviewer,
Thank you so much for your comments. As you recommended, we delete all references to biology or genetics. We included one study of your recommendation in the discussion.
We reviewed English language again with the help of a professional scientific translator (we include a certified of the review). We hope the manuscript could be accepted now. Thank you so much.

Reviewer 2 Report
An interesting analysis and it is good to see both consideration of the variable definitions of homelessness and the adoption of a specific, widely used definition of European homelessness in a population health paper. The comparison between migrant and non-migrant homeless population mortality risk adds a new dimension to this sort of analysis and is of wider interest as many Southern EU Member States have a similarly high level of migrant street/shelter homelessness.
Author Response
Dear reviewer,
Thank you so much for your comments.
We reviewed English language again with the help of a professional scientific translator (we include a certified of the review). We hope the manuscript could be accepted now. Thank you so much.

Reviewer 3 Report
Dear Authors,
you did a great job! However I think it could be useful to add a prospect on the possible developments of this research in terms of time frame. Indeed, you have analized 2006 data, quite old despite well collected.
It would be interesting to repeat this model on 2020 data, conditioneted by Covid-19 phenomenon in terms of scientific soundness of your work. Think about it!
Author Response
Dear reviewer,
Thank you so much for your response and your suggestions. I promise you that we accepted your suggestion, and we are planning a new phase of the research recruiting a new cohort of 2020. In the other hand we are following the same cohort from 2006 to 2020 and maybe we include data pf COVID in it to consider the influence of the pandemic in mortality. We include a new sentence before limitations to highlight these future perspectives. Thank you so much! Your words encourage us a lot to continue the hard work that is to investigate in Spain.